# The Effects of Assisted Freezing with Different Ultrasound Power Rates on the Quality and Flavor of Braised Beef

**DOI:** 10.3390/foods13101566

**Published:** 2024-05-17

**Authors:** Junguang Li, Chenhao Sun, Wuchao Ma, Kexin Wen, Yu Wang, Xiaonan Yue, Yuntao Wang, Yanhong Bai

**Affiliations:** 1College of Food and Bioengineering, Zhengzhou University of Light Industry, Zhengzhou 450001, China; 2014081@zzuli.edu.cn (J.L.); sunchenhao1010@163.com (C.S.); mawuchao66@163.com (W.M.); wenkexin202@163.com (K.W.); wy92@zzuli.edu.cn (Y.W.); 18338058683@163.com (X.Y.); 2017019@zzuli.edu.cn (Y.W.); 2Key Laboratory of Cold Chain Food Processing and Safety Control, Zhengzhou University of Light Industry, Ministry of Education, Zhengzhou 450001, China; 3Henan Food Laboratory of Zhongyuan, Zhengzhou University of Light Industry, Luohe 462000, China

**Keywords:** beef, ultrasonic–assisted immersion freezing, meat quality, flavor

## Abstract

This study investigated the effects of ultrasound–assisted immersion freezing (UIF) at different power rates (0, 200, 400, and 600 W) on the changes in beef quality and flavor after braising. The results demonstrated that UIF treatment at 400 W significantly reduced the juice loss (cooking loss decreased from 49.04% to 39.74%) and fat oxidation (TBARS value decreased from 0.32 mg/kg to 0.20 mg/kg) of braised beef. In addition, the tenderness (hardness value decreased from 5601.50 g to 2849.46 g) and color stability of braised beef were improved after UIF treatment. The flavor characteristics of braised beef were characterized using an electronic nose and an electronic tongue. The PCA analysis data showed that the cumulative contribution rates of the first and second principal components were 85% and 93.2%, respectively, with the first principal component accounting for a higher proportion. The UIF–400 W group had the highest concentration for the first principal component, and the differentiation was not significant compared to the control group. The total amino acid values of different power UIF treatment groups were improved compared to the AF treatment group, indicating that UIF can effectively reduce the losses caused by freezing. The results demonstrate that ultrasound–assisted freezing treatment is beneficial in enhancing the tenderness and flavor attributes of beef after braising, providing new insights into the processing of meat products with desirable quality characteristics.

## 1. Introduction

Freezing can inhibit the growth and reproduction of microorganisms in meat products, slow down the biochemical processes, and thus extend the shelf life [1]. However, traditional freezing methods such as air freezing and immersion freezing produce large and irregular extracellular ice crystals, resulting in muscle fiber rupture and reduced water retention, thereby reducing the quality of meat products [2,3,4]. The rate of the freezing process can affect the size and distribution of ice crystals, thus affecting the quality of frozen meat products [5]. Ultrasonic–assisted immersion freezing (UIF) is a new rapid freezing technology that combines ultrasound and immersion freezing and has been widely studied in assisted freezing of meat products in recent years. Zhang et al. [6] found that UIF treatment of beef can produce uniform and delicate ice crystals, which can effectively maintain the integrity of muscle tissue during frozen storage. They also found that ultrasound can reduce water migration and lipid oxidation during frozen storage, improving meat quality [7]. In addition, ultrasound can promote heat transfer and increase freezing speed. Sun et al. [8] found that ultrasound can shorten the freezing time of carp and significantly reduce thawing loss and braising loss. Therefore, ultrasound is a novel technology with great potential in assisting frozen storage of meat quality.

Braising in soy sauce is a traditional way of processing meat products in China. As one of the typical seasoned meat products in China, braised beef is usually treated with various spices and seasonings to enhance its taste. Meat braised for a long time will have a unique aroma and taste [9]. However, China’s prepared meat products face many problems, such as a high juice loss rate, poor color, and serious flavor damage [10]. Frozen storage is an important step before the subsequent processing of food, and freezing has a significant impact on the quality and flavor of braised meat [11,12,13]. Ultrasonic–assisted freezing propagates through a liquid medium, resulting in cavitation effects and microfluidization, which will destroy the muscle fiber skeleton, improve heat and mass transfer, and promote the mutual penetration of salt and other substances in meat and soup, thus effectively improving the overall flavor of meat. Kang et al. [14] found that ultrasound could reduce the hardness and chewability of beef and affect the sensory acceptability of meat after hot processing. At the same time, they have also shown that by freezing and storing fresh meat before braising and processing, the texture, color, and flavor of stewed meat improves [15]. At present, there have been many studies on the effects of low–temperature freezing on the quality and flavor of meat after processing [16,17,18,19], but there are few reports on the effects of different freezing methods on the edible quality and flavor of meat after braising.

The purpose of this study was to investigate ultrasonic–assisted immersion freezing (UIF) with different power rates, immersion freezing (IF), and air freezing (AF) on the moisture content, cooking loss, color, texture properties, and fat oxidation of braised beef. In addition, the flavor characteristics of braised beef were characterized using an electronic nose and an electronic tongue. The research results can provide theoretical guidance for meat processing.

## 2. Materials and Methods

### 2.1. Materials

Eyeround (mainly the semitendinosus muscle) was purchased from a Dennis supermarket (Zhengzhou, Henan, China), and the samples were kept at 0–4 °C and transported to the laboratory within 15 min. Subsequently, after removing the visible fat and connective tissue around, it was divided into meat blocks 10 × 10 × 5 cm^3^ in size, with an average weight of 130 ± 5 g. Blocks of the beef were separately packed in polyethylene bags (Thickness 0.08 μm, Zhejiang Mingke Plastics Co., Ltd., Ningbo, China) and were stored in a refrigerator at 4 °C for 6 h to obtain the same initial temperature. All reagents were analytically pure.

### 2.2. Sample Preparation

All samples were divided into six treatment groups. Except for the control group, which was only refrigerated at 4 °C for 6 h, the other groups were subjected to air freezing (AF), immersion freezing (IF), and ultrasonic–assisted immersion freezing with different power rates (UIF power: 200 W, 400 W, and 600 W). AF was completed in a −20 ± 0.5 °C high and low temperature alternating chamber (BPHJ–500C, Shanghai Yiheng Technology Co., Ltd., Shanghai, China). IF and UIF were performed in ultrasound–assisted freezing equipment (SJT–2–10 L, Shangjia Biotechnology Co., Ltd., Shanghai, China). The operating frequency of the equipment was 20 kHz, the temperature was –20 ± 0.5 °C, and the power was continuously adjustable within the range of 50–2000 W. When frozen by IF, the sample was completely impregnated using a liquid medium (95% industrial alcohol) without ultrasound, while when frozen by UIF, ultrasound was turned on when the central temperature of the sample dropped to 0 °C, and the working mode was 5 s on /5 s off. All samples were frozen when the central temperature reached –18 °C. The temperature monitoring and recording of the freezing process were completed with a multichannel temperature inspection instrument (AT4508, Changzhou Applent Precision Instrument Co., Ltd. Changzhou, China).

After thawing the frozen sample at 4 °C, the surface–leached substances were washed clean with 4 °C water, and then the samples were marinated with sauce (taking 1 kg sample as an example, the sauce formula was 40 g of salt, 5 g of monosodium glutamate, 20 g of white sugar, and 4 g of composite phosphate) for 24 h. Subsequently, the sample and spices (using a 1 kg sample as an example, spices were formulated as 2.4 g of star anise, 1.3 g of dried chili peppers, 0.4 g of fennel, 0.2 g of Chinese prickly ash, 0.6 g of cinnamon, 0.4 g of fragrant leaves, and 1.2 g of red dates) were placed together in brine (using a 1 kg sample as an example, the brine formula comprised the amount of water that could completely immerse the sample, 171.3 g of light soy sauce, 51.4 g of sugar, 12 g of braising wine, 8.6 g of salt, 3.4 g of monosodium glutamate, 2.6 g of oil consumption, 94.6 g of onions, 5.1 g of scallions, and 4.1 g of ginger) that had been boiled in an induction cooker (temperature setting: 180 °C) (C21–SDHCB9E32, Shaoxing Supor Household Appliances Co., Ltd., Shaoxing, China) and braised for 2 h. Raw materials were purchased from Dennis Supermarket. After braising, all samples were naturally cooled and placed in a 4 °C refrigerator for testing purposes.

### 2.3. Moisture Content

A meat composition analyzer (FoodScan^TM^1, FOSS, Beijing, China) was used to determine the moisture content of the braised sample. The instrument was adjusted through a standard plate before the samples were tested. The braised beef chunks were cooled to room temperature and ground using a meat grinder (GRINDOMIX GM 200, Verder Shanghai Instruments and Equipment Co., Ltd., Shanghai, China). Thirty grams of braised beef were evenly spread on a dedicated sample tray of the meat composition analyzer for moisture content measurement. Determination conditions: Halogen lamp 35 W; scan data points 100; wavelength range 850–1050 nm. Each treatment group was measured three times.

### 2.4. Cooking Loss

The weight of each meat block in different treatment groups was expressed as m_0_. According to 2.2 sample preparation, the meat samples (130 ± 5 g) were braised for 2 h (180 °C) and cooled to room temperature. The surface moisture and exudates of the stewed samples were removed using filter paper, and the meat samples were weighed to m_1_. The experiment was repeated three times. The calculation formula for cooking loss was as follows:(1)$$\mathrm{Cooking}\;\mathrm{loss}(\%)=\frac{m_{0}–m_{1}}{m_{0}}\times 100$$

### 2.5. Color Determination

The sample was cooled to room temperature, and the surface moisture was absorbed using absorbent paper, and then color attributes were determined with a colorimeter equipped with a D65 light source and a 10° observer with a 10 mm diameter measuring area (WSC–80C, Beijing Optical Century Instrument Co., Ltd., Beijing, China). After the colorimeter was adjusted and calibrated with a black and white board, the samples from each treatment group were divided into three equally sized meat chunks (20 × 20 × 5 mm^3^) and tightly attached to the detector. The L* (lightness), a* (redness), and b* (yellowness) values of the sample were determined. Each treatment group was measured three times.

### 2.6. Texture Profile Analysis

The meat samples that had been braised and cooled to room temperature were subjected to texture profile analysis (TPA. XT. plus, Stable Micro Systems, Godalming, UK) according to Li et al. [20]. The sample meat was cut into small pieces of 20 mm × 20 mm × 10 mm along the direction of muscle fibers. Afterward, the texture was measured in the TPA mode (using P/50 probes). The measurement parameters include a measurement rate of 1 mm/s, a pre–measurement rate of 2 mm/s, a post–measurement rate of 2 mm/s, a residence time of 5 s for two compressions, and a compression ratio of 70% for the compression mode. 

### 2.7. Thiobarbituric Acid Reactive Substances (TBARSs)

The determination of TBARS values was carried out as described by Zhang et al. [21], with some modifications. Briefly, 5 g of the sample was homogenized (3 × 20 s, 9000 rpm, 4 °C) in 25 mL (*w*/*v*) 17.5% trichloroacetic acid (containing 0.1% ethylene diaminete traacetic acid). After filtration, 2 mL of the filtrate was uniformly mixed with thiobarbituric acid (2.0 mL, 0.02 M) and reacted in boiling water for 0.5 h. After cooling the mixture to room temperature (25 °C) and centrifuging for 5 min (10,000 rpm, 4 °C), the supernatant was mixed with 1 mL of chloroform and layered to obtain a new supernatant. The newly obtained supernatant was measured for absorbance at 532 and 600 nm. The results are expressed in milligrams of malondialdehyde (MDA) per kilogram of beef sample.

### 2.8. Electronic Nose

The volatile flavor of the sample was analyzed using an electronic nose (PEN3 electronic nose, Airsense, Germany), which consists of a sampling device, a sensor array detector, and Winmuster software (Version 1.6.2) for data recording and analysis. The sensor array includes 10 metal oxide semiconductor chemical sensors, namely W1C (1, sensitive to aromatic constituents, benzene), W5S (2, highly sensitive to nitrogen oxides), W3C (3, sensitive to ammonia), W6S (4, sensitive to hydrides), W5C (5, sensitive to olefin, short–chain aromatic compounds), W1S (6, sensitive to methyl), W1W (7, sensitive to sulfides, pyrazine), W2S (8, sensitive to alcohols, aldehydes, and ketones), W2W (9, sensitive to organic sulfides), and W3S (10, sensitive to long–chain alkanes) [22]. The electronic nose was measured according to Du et al. [23]. After removing the surface air–dried and wrinkled parts from each group of braised beef samples, they were cut into small pieces, and approximately 3 g was weighed. Then, each sample was sealed in an electronic nose–specific headspace bottle and left to stand for 30 min before conducting an electronic nose analysis of the headspace gas. Determination conditions: 25 °C, air as the carrier gas, cleaning time 120 s, preparation time 5 s, injection volume 60 mL/min, test time 480 s. To ensure the stability and accuracy of the experimental results, radar image analysis was performed using signals at 460 s, and the data within the 460–462 s were used for principal component analysis (PCA).

### 2.9. Electronic Tongue

The electronic tongue (cTongue, Shanghai Bosin Industrial Development Co., Ltd., Shanghai, China) was based on the electrode system as the main principle, equipped with six sensors corresponding to six precious metals (tungsten, gold, titanium, palladium, silver, and platinum) to detect the overall characteristic response signal of the sample, and the response signal was used for sample taste analysis [24]. Briefly, 50 g of minced meat and distilled water (in a ratio of 1:5) were added to a 250 mL beaker. After mixing, the beaker was placed in a 40 °C constant temperature water bath for 30 min. It was then filtered with 8 layers of gauze, and the filtrate was centrifuged at 3000× *g* for 20 min. The supernatant was stored at 4 °C for measurement. The testing steps were as follows: First, before sampling the electronic tongue sensor, in order to ensure a consistent amount of sample loading for each treatment group, 60 mL of filtrate sample was taken from each treatment group and poured into the loading beaker. Then, before and after each sample measurement, the sensor was electrochemically washed (that is, all electrodes were immersed in deionized water for 30 s). Finally, the sensor was washed by potential scanning, and each sample was retested eight times.

### 2.10. Amino Acid Analysis

Amino acid analysis was performed with reference to Oh et al. [25]. After each samp I have checked and revised all. le was crushed using a meat grinder, 20 mL of ultrapure water was thoroughly mixed with 5 g of the sample. Then, 20 mL of 10% 5–sulfosalicylic acid was added to the mixture and homogenized (3 × 20 s, 10,000 rpm, 4 °C) for deproteinization. After deproteinization, the homogenate was centrifuged at 15,000× *g* for 10 min, and the supernatant was collected. Then, 2 mL of n–hexane was added and shaken well, filtered through a 0.22 μm organic phase filter membrane, and finally measured with an automated amino acid analyzer (Biochrom 30+, Biochrom Ltd., Cambridge, UK). The amino acid analyzer conditions were as follows: precolumn, narrow pore, stainless steel, 30 mm; column, narrow pore, stainless steel, 125 mm, ID 3 mm; column temperature, 400 °C; injection volume, 20 μL; detector excitation wavelength, 440 nm; emission wavelength, 570 nm; program time, 87.50 min.

The sample determination solution was mixed with a volume of amino acid standard working solution (using AA–S–18), and then the mixture was injected into the amino acid analyzer [25]. Using the external standard method, the content of each amino acid in the sample was calculated using peak area. Amino acid standard solutions were used for identification and quantification, and free amino acids were expressed in milligrams per 100 g of meat (mg/100 g).

### 2.11. Statistical Analysis

Each sample was measured in triplicate, and the results are expressed as mean ± standard deviation. Single–factor analysis of variance (ANOVA) using SPSS statistics 25 was used to analyze water content, braising loss, tenderness, TBARS, and color data. Duncan’s multiple–comparison test was used for significance analysis (*p* < 0.05), and principal component analysis and data mapping were performed in Origin 2019.

## 3. Results and Analysis

### 3.1. The Effect of UIF on the Moisture Content of Braised Beef

The moisture content of different samples after braising is shown in Figure 1. It can be observed that the moisture content of all frozen samples after sauce brine treatment significantly decreased (*p* < 0.05), which was due to the severe contraction of muscle fibers in the meat samples during long–term high–temperature braising, weakening the binding force between muscle fibers and water, and leading to water loss [26]. However, the moisture content of UIF (59.90%) was higher than that of the AF (56.83%) and IF (57.58%) groups, and the UIF–400 W group had significantly higher values than that of the UIF–200 W (58.48%) and UIF–600 W (58.11%) groups, which were closest to the control group (60.17%), indicating that UIF can maintain the moisture content of the samples after sauce brine to a certain extent compared to traditional freezing methods. This may be due to the ultrasonic cavitation effect, resulting in changes in protein structure and function, increased solubility, improved ability to bind to water, and thus increased water content [27]. However, low ultrasound power had little effect on beef quality, while high ultrasound power could cause damage to muscle fibers, thereby weakening their ability to bind with water.

### 3.2. Effect of UIF on the Cooking Loss of Braised Beef

Cooking loss is an important indicator for evaluating the water–holding capacity of meat products, which has a significant impact on the yield of meat products. It includes the loss of some soluble substances in the meat sample, with water being the main component [28]. As shown in Figure 1, the cooking loss of the AF sample was larger and significantly different from that of other treatment groups (*p* > 0.05), with a negative correlation between cooking loss and water content in each treatment group. The moisture in the meat is maintained by the intact muscle structure, and the increase in cooking loss is mainly caused by heat–induced protein denaturation and muscle fiber protein contraction, causing damage to the muscle structure and ultimately leading to water loss [29]. Studies have shown that after AF, muscle fibers are twisted and deteriorate, and the slow freezing process forms large and uneven ice crystals, resulting in changes in the structure and functional characteristics of myofibrillar protein. The destruction of muscle structure and protein denaturation reduce the water–holding capacity of the sample, thereby increasing the water loss during cooking [30].

### 3.3. Thiobarbituric Acid Reactive Substances (TBARSs)

Figure 2 demonstrates the TBARS content of brined beef after different freezing treatments. The TBARS is an indicator of the degree of further oxidation of lipid primary oxidation products to secondary oxidation products in terms of the ratio of malondialdehyde (MDA) equivalents to the dry basis of the meat samples, which is important for accurately reflecting the degree of fat oxidation [31]. During the heating process, the fat of beef is oxidized to primary oxidation products, and then further heat is applied to produce malondialdehyde, a representative substance of secondary oxidation products in the lipid oxidation process, and higher values of malondialdehyde indicate poorer meat quality [32]. Proper fat oxidation can enhance flavor, but excessive fat oxidation can lead to spoilage of meat samples. The meat used in this experiment had low fat content, so the MDA values after oxidation were relatively small in all treatment groups. The results showed that the MDA values of all treatment groups were significantly higher than those of the control group (*p* < 0.05), apart from the UIF–400 W group, which could inhibit fat over–oxidation, with MDA similar to those of the control group. However, the difference between ultrasound treatments with 200 W and 600 W power was not significant, which might be due to the elevated TBARS content caused by the generation of more free radicals as a result of the excessively high cavitation effect produced at higher ultrasound power [33].

### 3.4. The Effect of UIF on the Color of Braised Beef

Color is closely related to the quality of deep–processed products, which will directly affect the acceptability of consumers. The results of the effect of different freezing methods on the surface color of braised beef with soy sauce are shown in Table 1. As indicated, the L* and a* values of each group treated with UIF were significantly increased compared to the AF and IF groups (*p* < 0.05), and the differences between the UIF groups were small. The increase in L* values may be due to the relatively high surface moisture content of beef, which increases light reflection, and these changes are influenced by muscle water–holding capacity [34]. The increase in a* value might be due to the ·OH generated after ultrasound treatment of the meat sample, which promoted lipid and protein oxidation, leading to the oxidative denaturation of myoglobin into oxymyoglobin [35]. There was no significant difference in b* values between all treatment groups, which might be due to the soy sauce used during the braising process imparting stable pigments to the meat sample. This result is slightly different from the research trend of Li et al. [36], possibly due to the longer braising time, which weakened the differences between different treatment groups.

### 3.5. Effect of UIF on the Texture of Braised Beef

Texture is one of the most important attributes that affect consumer acceptance, and the changes in the quality of braised beef are related to fat content and muscle fiber distortion, which have a significant impact on the sensory characteristics of the meat sample [37,38]. Table 2 shows the effects of different freezing methods on the texture characteristics of braised meat samples. The hardness of meat samples increased in all treatment groups compared to the control group, which was caused by a large amount of water loss from beef during prolonged high–temperature braising, the degradation of myofibrillar proteins, and the intense contraction of myofibers [39]. Among the groups, the hardness of AF samples was significantly higher than that of other groups (*p* < 0.05), which may be due to the severe distortion and deterioration of muscle fibers caused by slow freezing, leading to a decrease in the water–holding capacity of the samples [40]. After braising, the samples lost water severely. Therefore, as the hardness value increased, the tenderness decreased. However, it can be clearly observed that UIF–400 W treatment lowered the hardness of meat samples, which may be due to the cavitation and mechanical effects of ultrasound to promote the release of endogenous proteases and thus improve the tenderness of meat samples [34,41]. Too low ultrasound power had little effect on the meat sample, while too high ultrasound power actually caused a significant increase in the hardness of the meat sample. There was a significant difference in chewiness values between the treatment groups, which corresponded to hardness. For springiness and cohesiveness, an increase is observed in Table 2 for AF and IF, whereas all UIF treatment groups showed a decrease compared to the control group, but there was no significant difference, which was due to the loss of moisture and protein aggregation during the freeze–thaw process of the meat samples [42]. In conclusion, the hardness and chewiness of the UIF–400 W–treated meat samples after braising were closest to those of the control group, suggesting that the UIF–400 W treatment can improve the textural damage caused by the traditional freezing method.

### 3.6. The Effect of UIF on the Volatile Flavor Components of Braised Beef

An electronic nose is an electronic system that mimics the structure of the human nose to recognize specific molecules, which can distinguish the subtle changes in volatile substances [43]. It is an efficient, rapid, non–destructive, and environmentally friendly technical tool [23,44]. The PCA results of different samples are shown in Figure 3a. Different circles represent the signal collection points of volatile flavor substances in braised beef after different freezing methods [45]. The distance indicates the size of the odor differences between different meat samples, and the larger the distance, the larger the differences between the samples. However, the differences are considered to be relatively small due to the small contribution of PC2. As previously indicated, the contribution of the first and second principal components of the meat samples were 68.5% and 16.5%, respectively, with a cumulative total of 85%, indicating that these two principal components could reflect the characteristic information of the main flavor components of the meat samples. There were differences in the volatile matter composition of braised beef prepared with different freezing methods, and the overall odor distribution of the AF group differed most from that of fresh meat, while the UIF–400 W group was closest to the control group. The concentrations of different samples on the horizontal axis from small to large were AF, IF, UIF–600 W, UIF–200 W, UIF–400 W, and control, indicating that UIF had a certain improvement effect on the flavor concentration of the beef samples; the less overlapping the graphics of each processing group, the more open the differentiation of odors between different processing groups.

Figure 3b shows the radar graphs of volatile components in different treatment groups. As the figure illustrates, the shape of the radar graphs changed more obviously; in particular, the W1S, W1W, and W2W sensors were the most sensitive. This indicates that there were large differences in the composition of volatile compounds such as methyl analogs, inorganic sulfides, aromatic components, and organic sulfides in different samples, and the response value of the UIF–400 W group was larger than those of the other treatment groups, indicating that the flavor compounds in beef samples could be better preserved after the UIF–400 W treatment.

### 3.7. The Effect of UIF on the Taste of Braised Beef

The electronic tongue was developed to mimic the biological system of the human tongue to quickly distinguish different flavors of food [46]. The electronic tongue system can identify specific flavors using sensors that convert food chemicals into electrical signals, thus further analyzing the results [47]. An electronic tongue was used to detect the differences in flavor attributes of beef after braising using different freezing methods. The contribution of the first and second principal components of PCA was 86.9% and 6.3%, respectively, with a cumulative contribution of 93.2%, indicating that the analysis can accurately reflect the differences in the overall profile of the taste between the components; the further the distance between the graphs in the principal components, the better the differentiation of taste composition between the treatment groups. Figure 4 shows that the results of UIF–400 W were significantly different from the other treatment groups, but the separation distance from the control group was not obvious, which indicated that the taste compounds of the two groups were relatively similar, and the two groups accounted for a larger proportion of the horizontal coordinate, which indicated that the treatment of UIF–400 W improved the taste composition of beef after braising. However, the overall contour of the scatter plots of other treatment groups was far from these two groups, indicating that the flavors of other treatment groups were damaged to varying degrees.

### 3.8. Amino Acid Analysis

Amino acid is the basic unit of protein and an important index for evaluating meat quality [48]. Table 3 shows the free amino acid content in beef braised with control, AF, IF, UIF–200 W, UIF–400 W, and UIF–600 W treatments. A total of 16 amino acids were detected (9 essential amino acids and 7 non–essential amino acids), with a total essential amino acid content of 124.63 ± 0.52, 113.57 ± 0.18, 104.70 ± 0.01, 123.58 ± 0.15, 117.62 ± 0.12, and 113.57 ± 0.06 (mg/100 g). The total content of non–essential amino acids was 203.88 ± 0.84, 148.89 ± 0.09, 121.66 ± 0.20, 159.09 ± 0.12, 209.26 ± 0.30, and 153.06 ± 0.16 (mg/100 g) for control, AF, IF, UIF–200 W, UIF–400 W, and UIF–600 W treatments, respectively. As can be seen, the UIF–400 W treatment can effectively alleviate amino acid degradation. The total amino acid content of essential and non–essential amino acids in the UIF–400 W group was closest to that of the control group, and there was a significant difference between the treatment groups and the control group (*p* < 0.05). The quality of meat was also correlated with the total amount of amino acids; the higher the total amount of amino acids, the better the meat quality [49]. It has been proven that the aspartic (Asp) and glutamic (Glu) have a synergistic effect on the umami of meat [50,51]. The Glu content in the control and UIF–400 W groups was much higher than that in other treatment groups, and glutamic acid had the highest content among all amino acids, proving that UIF–400 W treatment can better preserve the umami of meat. Proline (Pro) and threonine (Thr) are also umami and sweet amino acids, which have a positive effect on the presentation of meat umami [52]. Isoleucine (Ile), leucine (Leu), and histidine (His), as typical bitter amino acids, can cause a bad flavor to meat samples, resulting in worse taste [53]. Methionine (Met) is the least abundant because it is temperature–sensitive and easily oxidized, indicating that it is most damaged during high–temperature braising [54]. Lysine (Lrg) may undergo a Maillard reaction with fatty acids, which produces dark substances [55,56]. Compared with the control group, the UIF–400 W group had the least Lrg loss, while other treatment groups had varying degrees of reduction, which was consistent with the color difference brightness value. In summary, the significant differences in amino acid composition and content of meat samples between different freezing methods are shown in Table 3. UIF–400 W treatment could effectively reduce the losses caused by freezing.

## 4. Conclusions

In this study, the effects of air freezing, dip freezing, and ultrasonic–assisted dip freezing with different power rates on the quality of braised beef were investigated. The results showed that UIF inhibited the growth of ice crystals, better maintained the moisture content of the samples, and reduced cooking loss and fat oxidation. UIF slowed the deterioration of beef color and also slowed down the reaction or decomposition of most amino acids compared to the traditional freezing method. In addition, PCA plots were used to analyze the differences in the volatile flavor components measured with an electronic nose and electronic tongue and the overall flavor profiles of the treatment groups, which indicated that ultrasound–assisted impregnation freezing was effective. These findings indicate that ultrasound–assisted immersion freezing is an effective freezing method to improve the flavor of braised beef, and the best effect is achieved when the ultrasound power is 400 W. In conclusion, ultrasound–assisted immersion freezing can better maintain the quality and flavor of braised beef than air freezing and immersion freezing.

## Figures and Tables

**Figure 1 foods-13-01566-f001:**
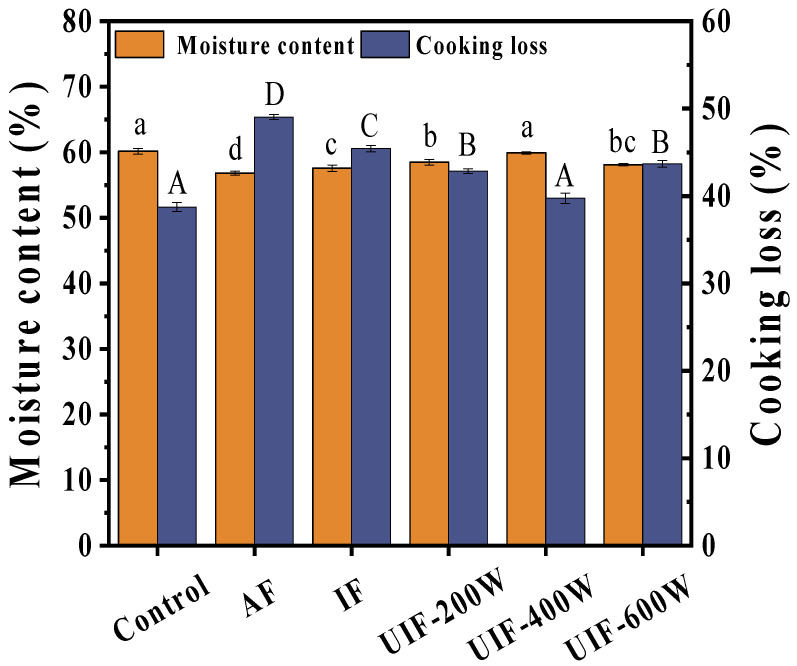
The moisture content and cooking loss of samples with different treatments. AF, air freezing; IF, immersion freezing; UIF–200 W, ultrasound–assisted immersion freezing at 200 W; UIF–400 W, ultrasound–assisted immersion freezing at 400 W; UIF–600 W, ultrasound–assisted immersion freezing at 600 W. Values are given as the means ± SD from triplicate determinations; ^a–d^ represents a significant difference in moisture content (*p* < 0.05); ^A–D^ represents a significant difference in cooking loss (*p* < 0.05).

**Figure 2 foods-13-01566-f002:**
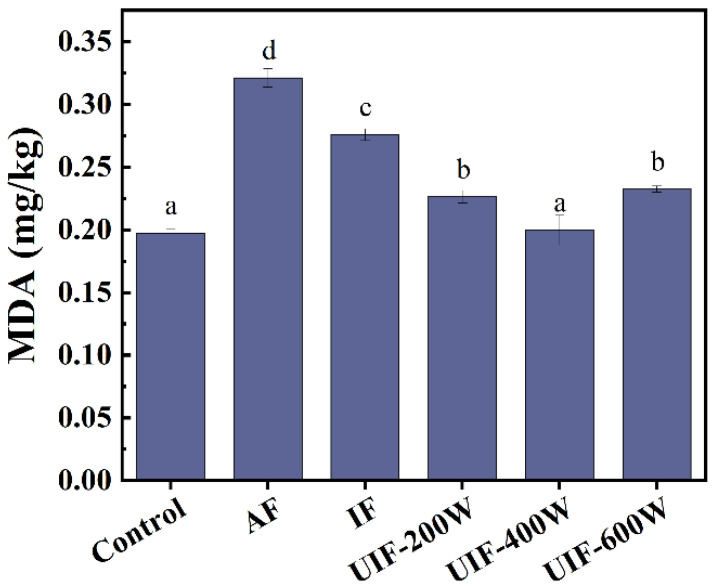
MDA content of samples with different treatments. AF, air freezing; IF, immersion freezing; UIF–200 W, ultrasound–assisted immersion freezing at 200 W; UIF–400 W, ultrasound–assisted immersion freezing at 400 W; UIF–600 W, ultrasound–assisted immersion freezing at 600 W. Values are given as the means ± SD from triplicate determinations. Different letters represent significant differences (*p* < 0.05).

**Figure 3 foods-13-01566-f003:**
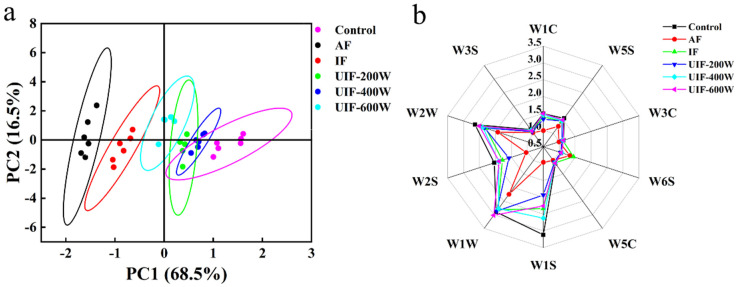
PCA (**a**) and radar (**b**) analysis of different treated samples. AF, air freezing; IF, immersion freezing; UIF–200 W, ultrasound–assisted immersion freezing at 200 W; UIF–400 W, ultrasound−assisted immersion freezing at 400 W; UIF–600 W, ultrasound–assisted immersion freezing at 600 W. Values are given as the value obtained by repeating three experiments.

**Figure 4 foods-13-01566-f004:**
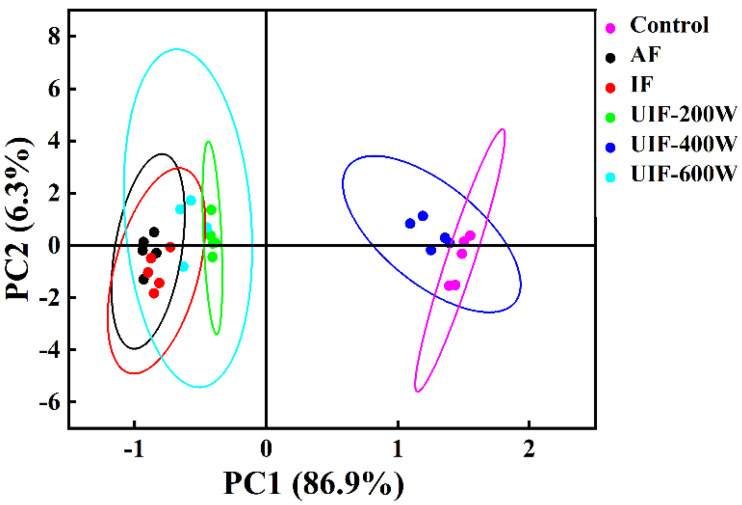
PCA analysis of the different treated samples. AF, air freezing; IF, immersion freezing; UIF–200 W, ultrasound–assisted immersion freezing at 200 W; UIF–400 W, ultrasound−assisted immersion freezing at 400 W; UIF–600 W, ultrasound–assisted immersion freezing at 600 W. Values are given as the value obtained by repeating three experiments.

**Table 1 foods-13-01566-t001:** Effects of different freezing methods on the color of samples after braising.

Group	L*	a*	b*
Control	47.78 ± 0.08 ^e^	9.86 ± 0.07 ^b^	11.14 ± 0.07 ^a^
AF	38.63 ± 0.08 ^a^	9.58 ± 0.16 ^a^	11.26 ± 0.14 ^ab^
IF	40.81 ± 0.11 ^b^	9.43 ± 0.06 ^a^	11.23 ± 0.08 ^ab^
UIF–200 W	46.91 ± 0.21 ^d^	10.00 ± 0.14 ^b^	11.31 ± 0.05 ^b^
UIF–400 W	47.67 ± 0.08 ^e^	9.84 ± 0.14 ^b^	11.14 ± 0.07 ^a^
UIF–600 W	46.42 ± 0.20 ^c^	9.95 ± 0.08 ^b^	11.23 ± 0.08 ^ab^

AF, air freezing; IF, immersion freezing; UIF–200 W, ultrasound–assisted immersion freezing at 200 W; UIF–400 W, ultrasound–assisted immersion freezing at 400 W; UIF–600 W, ultrasound–assisted immersion freezing at 600 W. Values are given as the means ± SD from triplicate determinations. Different letters in the same indicator indicate significant differences (*p* < 0.05).

**Table 2 foods-13-01566-t002:** Effect of different freezing methods on the texture of samples after braising.

	Control	AF	IF	UIF–200 W	UIF–400 W	UIF–600 W
Hardness (g)	1562.01 ± 75.85 ^e^	5601.50 ± 456.39 ^a^	4857.15 ± 106.49 ^ab^	3570.63 ± 301.95 ^cd^	2849.46 ± 35.42 ^d^	4197.88 ± 74.53 ^bc^
Springiness (cm)	0.52 ± 0.02 ^b^	0.67 ± 0.01 ^a^	0.66 ± 0.01 ^a^	0.63 ± 0.01 ^a^	0.55 ± 0.02 ^b^	0.62 ± 0.01 ^a^
Cohesiveness (N/mm^2^)	0.51 ± 0.03 ^b^	0.61 ± 0.03 ^a^	0.56 ± 0.01 ^ab^	0.56 ± 0.02 ^ab^	0.53 ± 0.01 ^ab^	0.56 ± 0.02 ^ab^
Gumminess	894.56 ± 30.84 ^d^	3184.33 ± 108.45 ^a^	2720.98 ± 97.01 ^ab^	1847.04 ± 114.27 ^c^	1528.44 ± 41.13 ^c^	2488.24 ± 67.12 ^b^
Chewiness	527.36 ± 36.24 ^e^	2142.04 ± 130.74 ^a^	1810.66 ± 78.79 ^ab^	1166.38 ± 123.44 ^cd^	802.16 ± 59.89 ^de^	1565.63 ± 21.21 ^bc^
Resilience	0.22 ± 0.02 ^a^	0.21 ± 0.01 ^a^	0.22 ± 0.01 ^a^	0.18 ± 0.02 ^a^	0.19 ± 0.01 ^a^	0.20 ± 0.01 ^a^

AF, air freezing; IF, immersion freezing; UIF–200 W, ultrasound–assisted immersion freezing at 200 W; UIF–400 W, ultrasound–assisted immersion freezing at 400 W; UIF–600 W, ultrasound–assisted immersion freezing at 600 W. Values are given as the means ± SD from triplicate determinations. Different letters in the same indicator indicate significant differences (*p* < 0.05).

**Table 3 foods-13-01566-t003:** The effect of different freezing methods on the amino acid content and TVA value of samples after braising.

Amino Acid (mg/100 g Meat)	Different Processing Groups						Threshold (mg/100 g Meat)	TVA					
	Control	AF	IF	UIF–200 W	UIF–400 W	UIF–600 W		Control	AF	IF	UIF–200 W	UIF–400 W	UIF–600 W
Thr	12.21 ± 0.02 ^a^	9.71 ± 0.03 ^c^	8.54 ± 0.02 ^d^	7.92 ± 0.03 ^e^	11.83 ± 0.01 ^b^	11.83 ± 0.01 ^b^	260	0.05	0.04	0.03	0.03	0.05	0.05
Val	14.54 ± 0.01 ^a^	13.51 ± 0.02 ^b^	10.71 ± 0.02 ^f^	11.57 ± 0.01 ^d^	12.70 ± 0.01 ^c^	11.16 ± 0.04 ^e^	40	0.36	0.34	0.27	0.29	0.32	0.28
Met	6.85 ± 0.02 ^a^	6.61 ± 0.03 ^b^	5.32 ± 0.03 ^d^	6.66 ± 0.03 ^b^	4.64 ± 0.01 ^e^	6.35 ± 0.05 ^c^	30	0.23	0.22	0.18	0.22	0.15	0.21
Ile	8.72 ± 0.01 ^b^	5.73 ± 0.02 ^e^	5.58 ± 0.02 ^e^	17.13 ± 0.02 ^a^	8.12 ± 0.11 ^c^	7.35 ± 0.04 ^d^	90	0.10	0.06	0.06	0.19	0.09	0.08
Leu	14.28 ± 0.07 ^a^	11.59 ± 0.03 ^e^	9.00 ± 0.03 ^f^	13.24 ± 0.3 ^c^	13.63 ± 0.07 ^b^	12.74 ± 0.025 ^d^	190	0.08	0.06	0.05	0.07	0.07	0.07
Phe	12.43 ± 0.08 ^a^	8.44 ± 0.01 ^d^	10.13 ± 0.04 ^c^	11.22 ± 0.02 ^b^	12.28 ± 0.17 ^a^	10.95 ± 0.035 ^b^	90	0.14	0.09	0.11	0.12	0.13	0.12
His	37.04 ± 0.14 ^e^	45.22 ± 0.01 ^a^	44.26 ± 0.01 ^b^	44.15 ± 0.02 ^b^	41.80 ± 0.07 ^c^	40.51 ± 0.01 ^d^	20	1.86	2.26	2.21	2.21	2.09	2.03
Lys	9.78 ± 0.14 ^a^	6.95 ± 0.02 ^d^	5.82 ± 0.01 ^e^	7.13 ± 0.03 ^d^	8.41 ± 0.02 ^b^	7.84 ± 0.04 ^c^	50	0.20	0.14	0.12	0.14	0.17	0.16
Arg	8.28 ± 0.06 ^a^	5.76 ± 0.02 ^b^	5.34 ± 0.02 ^c^	4.62 ± 0.01 ^e^	4.22 ± 0.01 ^f^	5.15 ± 0.03 ^d^	50	0.17	0.11	0.11	0.09	0.08	0.10
ΣEAA	124.63 ± 0.52 ^a^	113.57 ± 0.18 ^c^	104.70 ± 0.01 ^d^	123.58 ± 0.15 ^a^	117.62 ± 0.12 ^b^	113.57 ± 0.06 ^c^							
Asp	10.17 ± 0.01 ^a^	6.27 ± 0.06 ^e^	5.69 ± 0.01 ^f^	8.48 ± 0.02 ^c^	7.53 ± 0.12 ^d^	8.85 ± 0.04 ^b^	100	0.10	0.06	0.06	0.08	0.08	0.09
Ser	14.74 ± 0.02 ^b^	12.71 ± 0.04 ^c^	14.78 ± 0.10 ^b^	10.62 ± 0.01 ^d^	14.64 ± 0.05 ^b^	17.00 ± 0.00 ^a^	150	0.10	0.08	0.10	0.07	0.10	0.11
Glu	118.6 ± 0.10 ^b^	73.50 ± 0.02 ^d^	40.75 ± 0.06 ^f^	84.56 ± 0.01 ^c^	129.55 ± 0.03 ^a^	67.90 ± 0.07 ^e^	30	3.99	2.45	1.36	2.82	4.32	2.26
Gly	9.87 ± 0.10 ^a^	9.14 ± 0.04 ^b^	7.23 ± 0.035 ^d^	8.17 ± 0.02 ^c^	9.85 ± 0.01 ^a^	9.64 ± 0.14 ^a^	130	0.07	0.07	0.06	0.06	0.08	0.08
Ala	19.74 ± 0.2 ^a^	16.34 ± 0.05 ^f^	17.19 ± 0.03 ^d^	16.67 ± 0.02 ^e^	17.95 ± 0.10 ^c^	18.76 ± 0.02 ^b^	60	0.33	0.27	0.28	0.28	0.29	0.31
Cys–s	22.73 ± 0.1 ^d^	23.19 ± 0.05 ^c^	25.23 ± 0.04 ^b^	22.67 ± 0.07 ^d^	22.83 ± 0.05 ^d^	26.39 ± 0.08 ^a^	–	–	–	–	–	–	–
Pro	8.32 ± 0.7 ^b^	7.75 ± 0.035 ^cd^	10.02 ± 0.05 ^a^	7.85 ± 0.02 ^c^	7.67 ± 0.31 ^d^	4.49 ± 0.01 ^e^	300	0.03	0.03	0.03	0.03	0.03	0.01
ΣNEAA	203.88 ± 0.84 ^b^	148.89 ± 0.09 ^e^	121.66 ± 0.20 ^f^	159.09 ± 0.12 ^c^	209.26 ± 0.30 ^a^	153.06 ± 0.16 ^d^							

AF, air freezing; IF, immersion freezing; UIF–200 W, ultrasound–assisted immersion freezing at 200 W; UIF–400 W, ultrasound–assisted immersion freezing at 400 W; UIF–600 W, ultrasound–assisted immersion freezing at 600 W. Values are given as the means ± SD from triplicate determinations. Different letters in the same indicator indicate significant differences (*p* < 0.05). ΣEAA, total essential amino acids; ΣNEAA, the total amount of non–essential amino acids; Thr, threonine; Val, valine; Met methionine; Ile, isoleucine; Leu, leucine; Phe, phenylalanine; His, histidine; Lys, lysine; Arg, arginine; Asp, aspartate; Ser, serine; Glu, glutamate; Gly, glycine; Ala, alanine; Cys–s, cysteine–s; Pro, proline.

## Data Availability

The original contributions presented in the study are included in the article, further inquiries can be directed to the corresponding author.

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
