# Peer review of "The Effects of Assisted Freezing with Different Ultrasound Power Rates on the Quality and Flavor of Braised Beef"

_foods, 2024, doi:10.3390/foods13101566_

Round 1
Reviewer 1 Report
Comments and Suggestions for Authors
The paper entitled “Effects of assisted freezing under different ultrasound powers on the quality and flavor of braised beef” presents an interesting study evaluating the effects of three freezing methods on the quality and flavour of braised beef. They show that ultrasonic-assisted immersion freezing is an effective freezing method for improving the flavour of braised beef, and that the best effect is obtained when the ultrasonic power is 400 W. Some aspects need to be clarified.
1.- Is the year 2002 written on the top of the page correct?
2.- Page 5. first two lines: What does the equipment do to calculate humidity? What physical quantity does it relate to humidity? Mass loss after heating at a given temperature? The white board has a calibration certificate? Maybe what was done was to adjust the equipment with the white board. Explain better.
3.- Section 2.4: How do you check that all the water has been removed?
4.- Section 2.5: The colorimeter is not calibrated, what they do is to adjust the colorimeter. Substitute calibrated for adjusted. To calibrate means to calculate correction and uncertainty.
5.- Section 2.7: Did you make a calibration line? What is the concentration of the standards? What parameters did the resulting line give? Why do you measure at two wavelengths?
6.- Paragraphs 2.9 and 2.10: Not understood. The authors do not write the procedures well.
7.- Section 2.10: Indicate concentrations of the standards used for each concentration level. The equation of the resulting straight line and the R2 of the line.
Reviewer 2 Report
Comments and Suggestions for Authors
Manuscript foods-2970515, entitled “Effects of assisted freezing under different ultrasound powers on the quality and flavor of braised beef”
Recommendation: The above paper is not suitable in its present form.
The present article provides useful information about the effects of assisted freezing under different ultrasound powers on the quality and flavor of braised beef. It is in general appropriately organized, carried out and written, however there are some points that should be corrected or clarified.
· In abstract, how did you reach to the conclusion that “The total amino acid values of the UIF-400 W group were also closest to the control group, indicating that UIF can effectively reduce the losses caused by freezing”? Table 3 generally shows significant differences between UF-400 W and control group.
· In several parts in Material and methods, authors use imperative and not indicative (i.e. second paragraph of sample preparation, moisture content, electronic tongue)
· In Figure 1, please explain what is the meaning of superscripts (capital and lowercase)
Abstract
TBARS are measured in %?
P3
L2: “biochemical processes”
L5: “products [2-4]. The rate of freezing process can affect the size…”
L6: Please delete “the growth of ice crystals in the freezing process and”
L17: “…is usually treated with various spices…”
L19: “…meat products face many…”
P4
Materials
L5: Freezing or refrigerated temperature? Please specify the reagents
Sample preparation
L1-2: “…groups were subjected to air freezing…”
L8: “impregnated in the refrigerant”?
L14: “it was soaked” instead of “soak”
L18: “…spices were formulated as…”
P5
Color determination
L2: “…and then color attributes were determined…”
L4: “fastened to the mirror mouth”?
L5: “assessed” instead of “read”
Texture profile analysis
L1: “remained” instead of “aired”
Thiobarbituric acid reactive substances (TBARS)
L1-2: “The determination of TBARS values has carried out as described by Zhang et al. [21] with some modifications.”
P6
Electronic nose
LL9: “according” instead of “referred”
L10-11: “…small pieces and approximately 3 g were weighed. Then, each sample was sealed…”
Amino acid analysis
L1: “was” instead of “is”
L2: “were” instead of “is”
L5: “collected” instead of “taken”
P7
L4: Model of amino acid analyzer?
L5: Method? Please add a reference
Statistical analysis
L1: “Each sample was measured in triplicate, and the results are expressed…”
“3 Results and discussion”
The effect of UIF on the moisture content of braised beef
L1: “observed” instead of “seen”
L5-6: “However, the moisture content of UIF (59.90%) was higher than that of the AF (56.83%) and IF (57.58%) groups and the UIF-400W group had significantly higher values than that of the UIF-200W (58.48%)…”
P8
Effect of UIF on the cooking loss of braised beef
L2-3: “…of meat products. It includes the loss of some soluble substances…”
P9
L1: “low” instead of “less”
L4-5: “…(p<0.05), apart from UIF-400 W group that could inhibit fat over-oxidation, with MDA similar to those of the…”
The effect of UIF on the color of braised beef
L3: “As indicated” instead of “It can be seen that”
P10
Effect of UIF on the texture of braised beef
L12: “…clearly observed that UIF-400W treatment lowered the hardness of…”
L17: For springiness and cohesiveness and increase is shown in Table 2
P11
The Effect of UIF on the volatile flavor components of braised beef
L8: “As indicated” instead of “It can be seen that”
L19-20: “…treatment groups. As it is illustrated, the shape…”
L24-25: Please delete “The response value of the UIF-400 W group was significantly larger than that of other treatment groups,” (repetition)
P13
Amino acid analysis
L5-7: “…113.57 ± 0.06 (mg/100 g), and the total content of non essential amino acids was 203.88 ± 0.84, 148.89 ± 0.09, 121.66 ± 0.20, 159.09 ± 0.12, 209.26 ± 0.30, and 153.06 ± 0.16 (mg/100 g) for control, AF, IF, UIF-200 W, UIF-400 W, and UIF-600 W treatments, respectively. As shown, UIF-400W treatment…”
L8: Not for ΣEAA; UIF-200W was closer to control
L9: Please delete “other”
L11-12: “…meat quality [49]. It has been proven that the aspartic (Asp) and glutamic (Glu) have a synergistic…”
L13: Not for Asp!
L22: The least loss in what?
L24: “had no effect”? Significant differences are shown in Table 3
P14
Conclusions
What about lipid oxidation?
Comments on the Quality of English Language
Moderate editing of English language required
